# ReX-MLE: The Autonomous Agent Benchmark for Medical Imaging Challenges

**Roshan Kenia**[*1]                    ROSHAN_KENIA@FAS.HARVARD.EDU

**Xiaoman Zhang**[*1]                   XIAOMAN_ZHANG@HMS.HARVARD.EDU

**Pranav Rajpurkar**[1]

[1] *Department of Biomedical Informatics, Harvard Medical School, Boston, MA*

**Editors:** Accepted for publication at MIDL 2026

## Abstract

Autonomous coding agents built on large language models (LLMs) can now solve many general software and machine learning tasks, but they remain ineffective on complex, domain-specific scientific problems. Medical imaging is a particularly demanding domain, requiring long training cycles, high-dimensional data handling, and specialized preprocessing and validation pipelines, capabilities not fully measured in existing agent benchmarks. To address this gap, we introduce **ReX-MLE** , a benchmark of 20 challenges derived from high-impact medical imaging competitions spanning diverse modalities and task types. Unlike prior ML-agent benchmarks, ReX-MLE evaluates full end-to-end workflows, requiring agents to independently manage data preprocessing, model training, and submission under realistic compute and time constraints. Evaluating state-of-the-art agents (AIDE, ML-Master, R&D-Agent) with different LLM backends (GPT-5, Gemini, Claude), we observe a severe performance gap: most submissions rank in the 0th percentile compared to human experts. Failures stem from domain-knowledge and engineering limitations. ReX-MLE exposes these bottlenecks and provides a foundation for developing domain-aware autonomous AI systems.

**Keywords:** Autonomous Coding Agents, Medical Imaging, Benchmarking

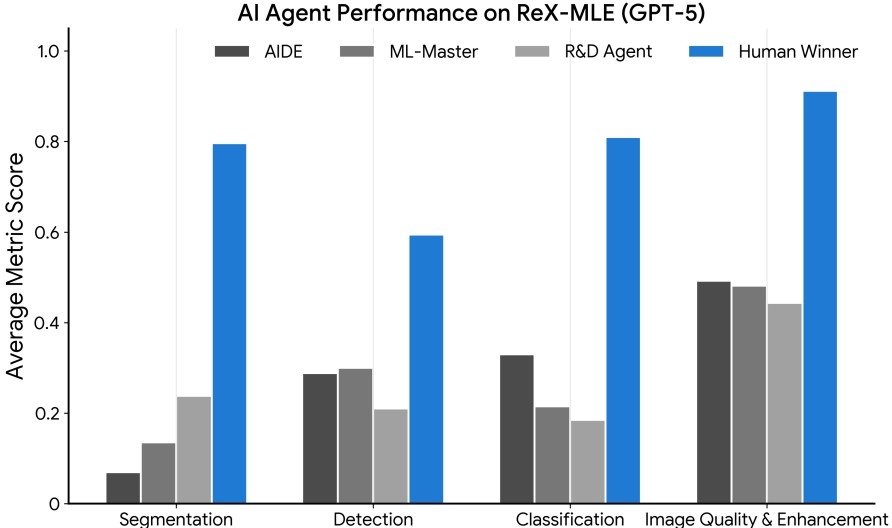

Figure 1: Performance of SOTA AI coding agents on ReX-MLE.

---

* Contributed equally

## 1. Introduction

Recent advances in large language models (LLMs) have enabled autonomous coding agents capable of solving standard machine learning engineering (MLE) and software engineering tasks (Liu et al., 2025; Chan et al., 2024; Yang et al., 2025b). These frameworks indicate the potential for agents to operate as autonomous AI scientists in the future (Gottweis et al., 2025). However, despite their success on general benchmarks, current agents break down when faced with complex, domain-specific scientific challenges (Zhu et al., 2025). Medical imaging represents a particularly demanding domain that exposes these limitations.

Unlike standard computer vision benchmarks, medical imaging tasks involve high-dimensional and heterogeneous modalities, including 3D CT and MRI volumes, multi-parametric scans, and gigapixel pathology slices, which require specialized preprocessing, normalization, and augmentation strategies (Shen et al., 2017). These processes often require the specialized insight of someone with substantial hands-on experience. Training these models often requires multi-stage pipelines, long training cycles, and careful hyperparameter tuning, demanding meticulous workflow management (Eisenmann et al., 2023).

Despite rapid progress in agent capabilities, existing evaluation frameworks remain poorly aligned with real-world scientific requirements. Current coding benchmarks (Jimenez et al., 2023; Chan et al., 2024; Tang et al., 2023; Padigela et al., 2025) tend to emphasize code generation or simplified ML pipeline construction, but overlook competition-grade metrics, assessment of actual model outputs under realistic training constraints, adherence to domain-expert preprocessing and validation standards, and comparison against documented winning strategies. Lacking these elements, prior benchmarks fail to surface the behaviors that prevent current agents from succeeding on real medical imaging tasks, such as early training termination, mismanagement of computational budgets, flawed preprocessing pipelines, and invalid evaluation procedures.

To address these gaps, we construct **ReX-MLE**, a benchmark comprised of **20** diverse challenge tasks derived from **10** high-impact medical imaging competitions. ReX-MLE spans **8** imaging modalities and multiple task types, including segmentation, detection, classification, image quality assessment, and generative enhancement. Within this environment, agents must independently handle the full scientific workflow, like preprocessing, model design, training, and evaluation, and are required to generate full submission-ready predictions (e.g., NIfTI volumes, masks, and JSON detection files) rather than single textual responses.

Our main contributions are: (i) We introduce ReX-MLE, a comprehensive benchmark for evaluating autonomous AI agents on domain-specific scientific challenges, derived from high-impact medical imaging competitions spanning segmentation, detection, classification, and generation. (ii) We establish a rigorous evaluation protocol that assesses agents' abilities to generate full, submission-ready prediction files under strict computational and time constraints, mirroring the requirements of real-world scientific discovery. (iii) We conduct an extensive evaluation of leading autonomous systems, including ML-Master and AIDE, utilizing state-of-the-art foundation models (GPT-5, Gemini 3 Pro, and Claude 4.5 Sonnet). As shown in Figure 1, **our results reveal a significant performance disparity, with even the most advanced agents consistently ranking in the lowest percentiles relative to human experts, highlighting critical gaps in domain-specific engineering and process validity.**

Table 1: Overview of the 20 ReX-MLE challenges across 10 competitions.

| Challenge | Modality | Task Type |
|---|---|---|
| USenhance (Guo et al., 2023) | Ultrasound | Image Generation |
| LDCTIQA2023 (Lee et al., 2025) | CT | Image Quality Assessment |
| PANTHER-T1 (Betancourt Tarifa et al., 2025) | MRI | Tumor Segmentation |
| PANTHER-T2 (Betancourt Tarifa et al., 2025) | MRI | Tumor Segmentation |
| SEG.A (Jin et al., 2025) | CTA | Segmentation |
| DENTEX (Hamamci et al., 2023) | X-ray | Detection |
| TopCoW-CTA-Seg (Yang et al., 2025a) | CTA | Segmentation |
| TopCoW-CTA-Det (Yang et al., 2025a) | CTA | Detection |
| TopCoW-CTA-Cls (Yang et al., 2025a) | CTA | Classification |
| TopCoW-MRA-Seg (Yang et al., 2025a) | MRA | Segmentation |
| TopCoW-MRA-Det (Yang et al., 2025a) | MRA | Detection |
| TopCoW-MRA-Cls (Yang et al., 2025a) | MRA | Classification |
| PUMA-Track1-TissueSeg (Schuiveling et al., 2025) | Pathology | Segmentation |
| PUMA-Track1-NucleiDet (Schuiveling et al., 2025) | Pathology | Nuclei Detection |
| PUMA-Track2-TissueSeg (Schuiveling et al., 2025) | Pathology | Segmentation |
| PUMA-Track2-NucleiDet (Schuiveling et al., 2025) | Pathology | Nuclei Detection |
| ISLES'22 (Hernandez Petzsche et al., 2022) | MRI | Segmentation |
| CellSeg (Ma et al., 2024) | Microscopy | Segmentation |
| TopBrain-CTA-Seg (Yang et al., 2025a) | CTA | Segmentation |
| TopBrain-MRA-Seg (Yang et al., 2025a) | MRA | Segmentation |

## 2. ReX-MLE

### 2.1. Benchmark Design

To systematically evaluate AI agent capabilities on medical imaging tasks, we construct **ReX-MLE**, a curated benchmark of medical imaging competitions adapted from Grand Challenge[1]. Our benchmark is designed to cover diverse modalities, task types, and clinical applications while maintaining rigorous standards for data quality and evaluation protocols, enabling systematic comparison of agent performance against expert-built solutions.

**Challenge Selection Criteria.** We select Grand Challenge tasks that meet standard requirements for data quality and reproducibility. Eligible challenges required substantial community participation (typically over 200 registrants or extensive submissions), open licensing that permits research use, and publicly released evaluation metrics and scripts. We included challenges with complete competition materials, including task descriptions and sample submissions. We ensured broad modality coverage and task diversity, spanning classification, detection, segmentation, regression, and image enhancement.

**Benchmark Composition.** ReX-MLE comprises 20 challenges across 10 major medical imaging competitions, covering diverse modalities, task types, and anatomical regions. Table 1 summarizes the benchmark composition. The dataset distribution requires agents to operate across highly heterogeneous data formats and clinical contexts, covering neurovascular imaging (TopCoW, SEG.A, ISLES'22, TopBrain), oncology (PANTHER, PUMA), microscopy (CellSeg), dentistry (DENTEX), and image quality enhancement and reconstruction (LDCT-

---

1. https://grand-challenge.org/challenges/

Figure 2: Overview of ReX-MLE Task Categories. This figure illustrates the four distinct task types included in the benchmark: Segmentation, Detection, Classification, and Image Generation.

IQA, USenhance). This diversity enforces generalization across anatomical structures and clinical objectives, providing a realistic testbed for autonomous medical model development.

**Challenge Adaptation.** For each challenge, we prepare standardized materials following the MLE-bench framework (Chan et al., 2024). Provided materials include a concise competition description, automated dataset-preparation scripts, sample submissions, and Python implementations of official evaluation metrics for fully local validation. Each task also includes a YAML metadata file defining challenge identifiers, task types, data paths, and grading parameters. All preparation and evaluation code is publicly released to support reproducibility and future extensions.

## 2.2. Baseline Agents

**AIDE** (Schmidt et al., 2024) frames machine learning engineering as a code-space optimization problem and performs greedy tree search with iterative refinement. The agent proposes

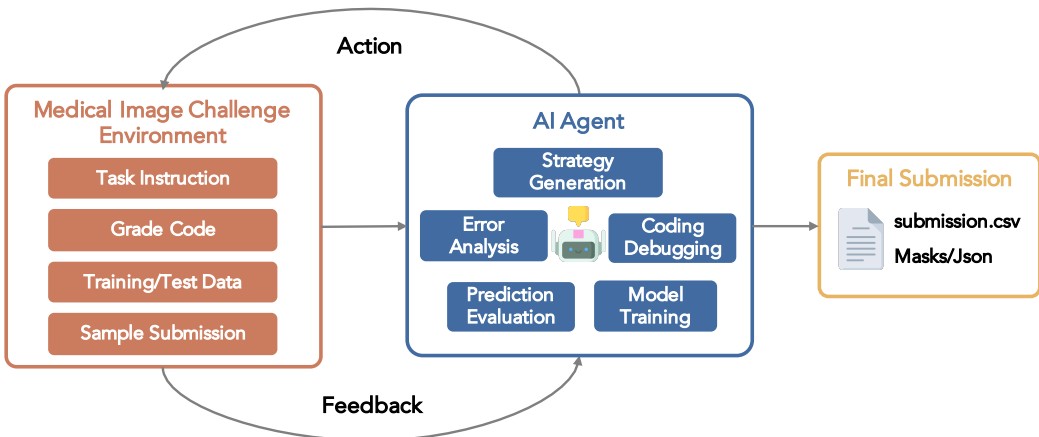

Figure 3: Autonomous Agent Interaction Workflow. This diagram depicts the workflow between the Medical Image Challenge Environment and the AI Agent. The environment provides task instructions, data, and grading feedback, while the agent iteratively performs strategy generation, error analysis, coding, debugging, and model training to produce a final submission.

candidate solutions, executes them, and uses automated feedback to diagnose errors and incrementally correct code.

**ML-Master** (Liu et al., 2025) integrates large-scale exploration and reflective reasoning through a Monte Carlo Tree Search (MCTS) framework. By maintaining multiple solution trajectories and adaptively balancing exploration of new strategies with exploitation of promising ones, ML-Master efficiently navigates complex search spaces.

**R&D-Agent** (Yang et al., 2025b) employs a dual-agent design that separates conceptual exploration from implementation refinement. A Researcher module proposes strategic directions, while a Developer module resolves execution errors and iteratively improves the code, enabling diverse and resilient multi-trace exploration.

### 2.3. Experimental Setup

**Computational Resources.** All agents are executed on standardized hardware consisting of NVIDIA H100 GPUs (80GB memory), 64 CPU cores, and 128GB RAM. This configuration reflects realistic research computing environments while remaining broadly accessible to the academic community. We select GPT-5 as the primary model for all experiments (unless otherwise noted) due to its widespread adoption and API accessibility.

**Time Budget.** Each agent is allocated a strict 24-hour wall-clock budget per challenge to develop a complete solution. This constraint evaluates the agent's ability to efficiently explore solution spaces, manage computational resources, and prioritize promising methodological directions under realistic time limitations.

**Agent Inputs.** For every challenge, agents receive five standardized inputs: (1) a competition description document outlining task objectives and clinical context; (2) the full

training dataset with accompanying annotations; (3) the test data without ground truth annotations; (4) a sample submission illustrating the expected output format; (5) a local evaluation script enabling iterative validation. The description follows a standardized format that includes: overview and clinical context, detailed task specifications (modality, organs, dataset characteristics), dataset organization and splits, evaluation metrics with mathematical formulations, and submission requirements. Without any human intervention, the agents must use these materials to autonomously perform the entire workflow, including data exploration, preprocessing, model design, training, validation, and final submission generation.

**Agent Outputs.** Each agent is instructed to produce a `submission.csv` file following the format of the provided sample submission. Unlike MLE-Bench, however, tasks that require saving predictions such as segmentations must be submitted as a folder containing both the CSV and the corresponding prediction files, rather than encoding all outputs within a single CSV. This setup reflects a realistic workflow in which model outputs need to be generated and used directly, rather than converted into easily represented text. We extract these submission folders for evaluation.

**Evaluation Metrics.** For each challenge, we collect the top 10 human competitor scores, or the maximum available if fewer, from the public test leaderboard. For each metric within a challenge (Appendix A), we reconstruct the competitors' positional rankings and then average these positions to obtain a mean ranking, which serves as the basis for each competitor's overall standing in the challenge. Although some competitors may have evaluated their submissions on private test sets, we recreate the evaluation conditions as closely as possible. We then assess agent performance comprehensively, using both absolute metrics and leaderboard-relative metrics tailored to the Grand Challenge environment:

- **Challenge-specific scores**: Raw performance on each challenge's evaluation metrics, as defined by their respective leaderboards. These scores allow for direct, like-for-like comparison across agents.
- **Competition rank**: For each challenge, we compute positional rankings for every metric relative to the human leaderboard, then average these positions to obtain a mean ranking ($\overline{\text{rank}}$). From this, we derive the agent's percentile using: $1 - \frac{\overline{\text{rank}}-1}{\text{number of human competitors}}$. Under this formulation for a competition with 10 human competitors, an agent with a mean ranking of 1 (best) is placed in the $100^{\text{th}}$ percentile, while a mean ranking of 11 (worst) corresponds to the $0^{\text{th}}$ percentile.
- **ReX-MLE Rank**: The overall benchmark ranking, computed as the mean of all competition percentiles across challenges (with failures assigned 0%).

## 2.4. Capability Evaluation Methodology

Beyond quantitative performance metrics, we conduct systematic capability analysis to understand *why* agents fail. For each agent and challenge, we evaluate the full execution logs, including reasoning traces, planning decisions, code generation, debugging attempts, and intermediate outputs, using the 13 "Winning Strategies" identified by Eisenmann et al. (2023) (Figure 4). This analysis is conducted through an automated LLM-as-a-judge pipeline, which applies a standardized rubric to determine whether there is explicit, verifiable evidence that an agent exhibited each strategy. A binary indicator is assigned accordingly, with

Table 2: Performance of autonomous ML agents on the medical subset of MLE-Bench. Each challenge is evaluated using its primary leaderboard metric (↑ indicates higher is better; ↓ indicates lower is better). Values are reported as raw metric scores. **Bold** text denotes the best agent score for each challenge.

| Challenge | Metric | Human | AIDE | | ML-Master | | R&D-Agent | |
|---|---|---|---|---|---|---|---|---|
| | | | w/o sol. | w/ sol. | w/o sol. | w/ sol. | w/o sol. | w/ sol. |
| HCD | AUROC ↑ | 0.983 | 0.988 | 0.989 | 0.990 | 0.992 | 0.981 | **0.993** |
| RANZCR | AUROC ↑ | 0.973 | 0.807 | 0.910 | 0.880 | 0.859 | **0.932** | 0.928 |
| ISIC | AUROC ↑ | 0.945 | 0.879 | **0.881** | 0.845 | 0.808 | 0.787 | 0.791 |
| HuBMAP | DICE ↑ | 0.948 | **0.133** | 0.000 | 0.000 | 0.000 | 0.008 | 0.000 |
| OSIC | LLL ↑ | -6.841 | -7.999 | -8.915 | **-7.389** | -8.191 | FAIL | -9.351 |
| UWGI | DICE/Haus. ↑ | 0.879 | 0.366 | 0.000 | 0.000 | 0.000 | **0.580** | 0.509 |
| RSNA-Spine | Combo ↓ | 0.276 | **0.563** | 0.761 | 0.600 | 0.675 | 4.959 | 0.656 |
| RSNA-BC | F1 ↑ | 0.490 | 0.056 | 0.048 | 0.028 | FAIL | 0.043 | **0.120** |
| RSNA-RG | AUROC ↑ | 0.600 | 0.454 | **0.547** | 0.500 | 0.462 | 0.540 | 0.480 |
| SIIM-COVID | mAP ↑ | 0.623 | 0.236 | **0.393** | FAIL | 0.305 | FAIL | FAIL |
| VinBig | COCO ↑ | 0.289 | FAIL | FAIL | FAIL | FAIL | **0.116** | FAIL |

detailed procedures and the exact prompts provided in Appendix B. Aggregating these scores across all 20 challenges yields capability profiles that reveal which core scientific and engineering practices current autonomous ML systems consistently fail to demonstrate.

## 3. Results

### 3.1. Performance on the Medical Subset of MLE-Bench

To establish a performance baseline, we first evaluated AIDE, ML-Master, and R&D-Agent on the medical subset of the original MLE-Bench. Table 2 shows that agents perform well on simpler tasks such as the HCD (Histopathologic Cancer Detection) challenge, where ML-Master achieved an AUROC of 0.992, surpassing the human gold standard of 0.983. However, across more complex medical imaging tasks, especially those involving segmentation or detection (e.g., HuBMAP, UWGI), performance deteriorated sharply. Multiple agents produced near-zero Dice or mAP scores despite correct training scripts, indicating fundamental deficiencies in handling high-dimensional medical image data.

We conducted an additional experiment on the medical subset of MLE-Bench in which we provided the agents with the winning solution reports created by the 1st-place winners of each challenge, excluding HCD, which did not have a report available. These reports described the methodology used by the winning teams but did not include the full training or evaluation code. Our analysis in Table 2 shows that the performance gap remains even when agents are explicitly given the top-performing strategy ("w/ sol."). The inability of agents to reproduce expert-level performance, even when provided with the solution, suggests that the limitation goes beyond domain knowledge or hypothesis generation and reflects a fundamental inability to carry out the engineering work needed to adapt, debug, and deploy medical imaging pipelines.

Table 3: Agent performance across the ReX-MLE suite with primary metric values and percentile ranks (Competition Rank) separated.

| Challenge | Metric (↑) | AIDE | | ML-Master | | R&D-Agent | | Human |
|---|---|---|---|---|---|---|---|---|
| | | Value | Rank | Value | Rank | Value | Rank | Value |
| *Segmentation Tasks* | | | | | | | | |
| ISLES'22 | Dice | 0.04 | 0% | 0.00 | 0% | 0.02 | 0% | 0.79 |
| NeurIPS-CellSeg | F1 | 0.04 | 0% | 0.04 | 0% | 0.36 | 0% | 0.88 |
| PANTHER-T1 | Dice | 0.33 | 16% | 0.13 | 8% | 0.16 | 8% | 0.73 |
| PANTHER-T2 | Dice | 0.09 | 10% | 0.05 | 8% | 0.28 | 58% | 0.53 |
| PUMA-T1-Seg | Dice | FAIL | – | 0.00 | 0% | 0.00 | 0% | 0.78 |
| PUMA-T2-Seg | Dice | 0.00 | 0% | 0.00 | 0% | 0.00 | 0% | 0.78 |
| SEG.A | Dice | 0.02 | 0% | 0.02 | 0% | 0.00 | 0% | 0.92 |
| TopBrain-CTA | Mean Dice | 0.03 | 2% | 0.26 | 3% | 0.08 | 2% | 0.79 |
| TopBrain-MRA | Mean Dice | 0.01 | 10% | 0.26 | 0% | 0.50 | 0% | 0.81 |
| TopCoW-CTA-Seg | Mean Dice | 0.09 | 0% | 0.25 | 3% | 0.49 | 2% | 0.87 |
| TopCoW-MRA-Seg | Mean Dice | 0.11 | 0% | 0.48 | 0% | 0.73 | 3% | 0.88 |
| *Detection Tasks* | | | | | | | | |
| DENTEX | AP | 0.09 | 0% | 0.08 | 0% | 0.09 | 0% | 0.40 |
| PUMA-T1-Det | F1 | 0.02 | 0% | 0.08 | 0% | 0.06 | 0% | 0.66 |
| PUMA-T2-Det | F1 | FAIL | – | 0.00 | 0% | 0.01 | 0% | 0.27 |
| TopCoW-CTA-Det | IoU | 0.67 | 38% | 0.65 | 25% | 0.70 | 56% | 0.79 |
| TopCoW-MRA-Det | IoU | 0.66 | 14% | 0.69 | 14% | 0.19 | 14% | 0.85 |
| *Classification Tasks* | | | | | | | | |
| TopCoW-CTA-Cls | Accuracy | 0.33 | 33% | 0.10 | 0% | 0.28 | 50% | 0.73 |
| TopCoW-MRA-Cls | Accuracy | 0.33 | 25% | 0.33 | 25% | 0.09 | 0% | 0.89 |
| *Image Quality & Enhancement Tasks* | | | | | | | | |
| LDCT-IQA | Score | 2.62 | 33% | 2.50 | 0% | 2.66 | 50% | 2.74 |
| USenhance | LNCC | 0.11 | 0% | 0.13 | 0% | FAIL | – | 0.91 |
| Overall Mean Percentile | – | – | 9.05% | – | 4.53% | – | **12.15%** | – |

## 3.2. Quantitative Performance Gap on ReX-MLE

We further evaluated AIDE, ML-Master, and R&D-Agent across all 20 challenges in ReX-MLE. As detailed in Table 3, agents failed to achieve expert-level performance across all 20 tasks, with the majority of submissions scoring in the $0^{th}$ relative to human competitors.

**Segmentation Tasks.** Segmentation represents the most challenging category. In pathology segmentation (PUMA), AIDE failed to produce valid submissions, and ML-Master and R&D-Agent achieved Dice scores of zero. These failures likely stem from agents' inability to handle gigapixel WSI data, whether via memory-efficient patching or proper preprocessing. Similar trends are observed in volumetric neurovascular tasks (TopBrain, TopCoW), where the majority of mean Dice scores remain below 0.3, despite winners exceeding 0.85. R&D-Agent shows modest improvement on certain TopCoW and TopBrain tasks, achieving Dice scores up to 0.73, though still below human performance. These results suggest difficulties in handling 3D spatial consistency, voxel spacing normalization, and robust volumetric inference.

**Detection and Classification Tasks.** Compared with segmentation, agents achieved slightly better performance on detection and classification tasks, but results remain substantially worse than human competitors. For example, on TopCoW-CTA-Det, R&D-Agent reached an IoU of 0.70 (56%), representing one of the few non-zero-percentile outcomes across the entire benchmark. Similarly, agents achieved moderate accuracies in the TopCoW classification tasks (e.g., AIDE at 33%), although still far below the winning solutions (0.73). Nevertheless, the overall performance remains poor: scores are inconsistent, often collapse to 0%, and never approach competitive human baselines. Thus, even on the tasks where agents show their best results, they are still far from demonstrating reliable or clinically meaningful competence.

**Generative and Quality Assessment.** Performance on generative tasks was notably poor, with the USenhance (Ultrasound enhancement) task showing agents achieved scores of $\sim 0.11$ against a human baseline of 0.91. Visual inspection suggests agents treated the task as simple style transfer without accounting for the specific speckle noise statistics of ultrasound physics. Conversely, on LDCT-IQA (CT Image Quality Assessment), R&D-Agent achieved a score of 2.66 (50th percentile), coming close to the winning score of 2.74. However, this strong performance may be influenced by the metric's sensitivity to global image statistics rather than a genuine understanding of diagnostic image quality, a pattern examined further in our failure taxonomy. Notably, LDCT-IQA is the only challenge that does not require submitting separate prediction files beyond a `submission.csv`, suggesting that these agents perform best when the submission process is substantially simplified but not realistic.

### 3.3. Capability Analysis

To investigate the underlying causes of the performance gap, we moved beyond outcome-level metrics and examined the process validity of each agent's workflow by evaluating every execution trace excluding code against the 13 Winning Strategies identified by Eisenmann et al. (2023) in their meta-analysis of biomedical competition winners. Given the scale of our benchmark, which includes 60 execution traces across 20 challenges, manual annotation was impractical, so we developed an automated adjudication pipeline powered by a state-of-the-art large language model (GPT-5) acting as a technical evaluator. This system parses full execution logs, including shell commands, Python code, and internal reasoning traces, and assigns each strategy a binary score: 1 when explicit evidence is present (for example, importing nibabel for resampling or applying test-time augmentation) and 0 when the evidence is absent or ambiguous. To validate this automated approach, we conducted human expert annotation on 50 randomly sampled traces. Two domain experts independently annotated whether each strategy was implemented, yielding 100% agreement with GPT-5's judgments. This high reliability stems from our deliberate reformulation of capability assessment as binary evidence detection under a strict, predefined rubric, rather than open-ended qualitative judgment.

The capability scores in Figure 4 reveal substantial differences in how agents leverage winning strategies. R&D-Agent demonstrates notably higher coverage across multiple strategies, particularly excelling in having domain knowledge (0.663), reflecting metrics in method design (0.638), postprocessing results (0.614), and analyzing and handling failure cases (0.601). In contrast, AIDE and ML-Master show comparable but considerably lower

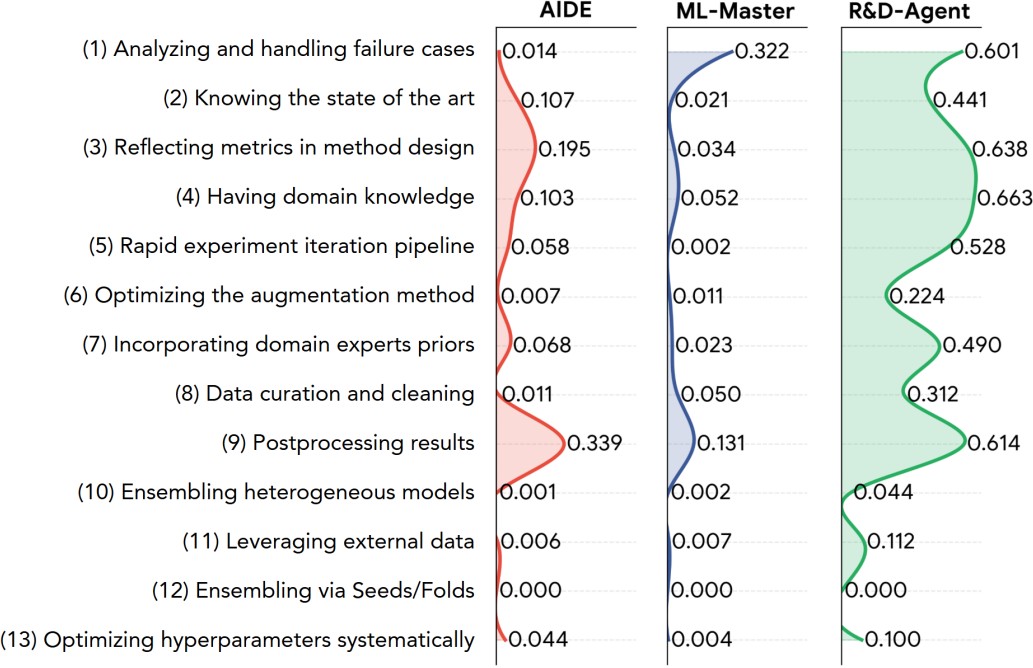

Figure 4: Comparison of ML research agent capabilities across 13 key success factors.

Table 4: 3-Day agent performance across representative ReX-MLE challenges with primary metric values and percentile ranks (Competition Rank) separated.

| Challenge | Metric (↑) | AIDE | | ML-Master | | R&D-Agent | | Human |
|---|---|---|---|---|---|---|---|---|
| | | Value | Rank | Value | Rank | Value | Rank | Value |
| ISLES'22 | Dice | 0.18 | 0% | 0.36 | 0% | 0.04 | 25% | 0.79 |
| DENTEX | AP | 0.10 | 0% | 0.02 | 0% | 0.03 | 0% | 0.40 |
| TopCoW-CTA-Cls | Accuracy | 0.31 | 25% | 0.26 | 25% | 0.26 | 25% | 0.73 |
| LDCT-IQA | Score | 2.24 | 0% | 2.64 | 33% | 2.70 | 83% | 2.74 |
| Overall Mean Percentile | – | – | 6.25% | – | 14.5% | – | 33.25% | – |

coverage overall. ML-Master exhibits modest strengths in analyzing and handling failure cases (0.322) and knowing the state of the art (0.021), while AIDE shows its highest engagement in postprocessing results (0.339) and reflecting metrics in method design (0.195). Despite these differences, all three agents converge on notably poor performance in several critical areas: optimizing the augmentation method, ensembling, and leveraging external data, where scores approach or reach zero. Overall, the results show that while R&D-Agent engages with winning strategies substantially more frequently than AIDE and ML-Master, even this higher coverage falls short of expert-level rigor, and as demonstrated by our previous results, capability presence does not guarantee correct execution or improved outcomes.

### 3.4. Effect of Time Budget and Model Backend

To assess how agent performance scales with compute and model choice, we ran two ablations: (1) extending the time budget from 24 to 72 hours, and (2) evaluating agents across three LLMs (GPT-5, Gemini 3, Claude Sonnet 4.5). Both ablations use four representative challenges spanning segmentation, detection, classification, and image quality assessment.

**Three-Day Experiments.** Table 4 shows that extending the time budget to three full days yields minimal improvement for most agents and tasks. On ISLES'22 and DENTEX, all agents remain in the 0th percentile, except for R&D-Agent on ISLES'22, which reaches the 25th percentile. Classification performance on TopCoW-CTA-Cls improves modestly to the 25th percentile for all agents but remains far below expert solutions. The only substantial gain is observed on LDCT-IQA, where R&D-Agent reaches the 83rd percentile, suggesting that simpler, non-volumetric tasks benefit more from additional time than complex 3D tasks. Overall mean percentile ranks are 6.25% for AIDE, 14.5% for ML-Master, and 33.25% for R&D-Agent, indicating that increased time alone does not resolve core failure modes.

**Model Backend Ablation.** Table 5 evaluates AIDE and ML-Master using GPT-5, Gemini 3, and Claude Sonnet 4.5 as their backend LLMs. We exclude R&D-Agent from these results as it has no native backend support for Claude and Gemini. While LLM choice influences absolute scores, the overall pattern of failure is unchanged. In segmentation (ISLES'22), Claude produces the highest Dice score (0.65), yet still ranks in the 25th percentile, indicating that even different backends cannot compensate for missing domain-specific engineering. In detection (DENTEX), Gemini enables AIDE to reach the 8th percentile, but performance remains far below the winning 40% AP. Classification tasks show minor variation across backends but remain capped at 25–33% percentiles. For LDCT-IQA, backend differences again alter absolute scores but do not meaningfully enable agents to approach competitive performance. Across all tasks, backend effects are second-order relative to structural agent limitations: agents cannot reliably build, train, or validate domain-appropriate medical imaging pipelines, regardless of the foundation model driving their reasoning.

## 4. Related Work

**Benchmarks for Evaluating AI Agents.** Robust evaluation frameworks are essential for measuring AI agent capabilities across multiple dimensions. Code-focused benchmarks include HumanEval (Chen, 2021) for code generation, CRUXEval (Gu et al., 2024) for execution reasoning, and SWE-bench (Jimenez et al., 2023) for real-world GitHub issues. ML workflow benchmarks provide comprehensive end-to-end evaluation: MLE-bench (Chan et al., 2024) curates 75 Kaggle competitions revealing up to 50.67% medal rates for leading agents, while ML-Bench (Tang et al., 2023), ML-Dev-Bench (Padigela et al., 2025), and TimeSeriesGym (Cai et al., 2025) evaluate repository-level and time series tasks. Domain-specific reasoning benchmarks test specialized knowledge, with mathematics evaluations evolving from saturated benchmarks like GSM8K (Cobbe et al., 2021) and MATH (Hendrycks et al., 2021) to harder challenges including Putnam-AXIOM (Gulati et al., 2025) and MATH-Perturb (Huang et al., 2025), while LLM-SRBench (Shojaee et al., 2025) evaluates scientific equation discovery. However, existing benchmarks have critical limitations for evaluating

Table 5: Varying backend LLM performance across representative ReX-MLE challenges with primary metric values and percentile ranks (Competition Rank) separated. The specific models used are GPT-5, Gemini 3, and Claude Sonnet 4.5

| Challenge | Model | Metric ($\uparrow$) | AIDE | | ML-Master | | Human |
|---|---|---|---|---|---|---|---|
| | | | Value | Rank | Value | Rank | Value |
| ***Segmentation Tasks*** | | | | | | | |
| ISLES'22 | GPT | Dice | 0.04 | 0% | 0.00 | 0% | 0.79 |
| ISLES'22 | Gemini | Dice | FAIL | – | 0.46 | 25% | 0.79 |
| ISLES'22 | Claude | Dice | 0.45 | 5% | 0.65 | 25% | 0.79 |
| ***Detection Tasks*** | | | | | | | |
| DENTEX | GPT | AP | 0.09 | 0% | 0.08 | 0% | 0.40 |
| DENTEX | Gemini | AP | 0.20 | 8% | 0.19 | 1% | 0.40 |
| DENTEX | Claude | AP | 0.02 | 0% | 0.12 | 0% | 0.40 |
| ***Classification Tasks*** | | | | | | | |
| TopCoW-CTA-Cls | GPT | Accuracy | 0.33 | 33% | 0.10 | 0% | 0.73 |
| TopCoW-CTA-Cls | Gemini | Accuracy | 0.35 | 25% | 0.33 | 25% | 0.73 |
| TopCoW-CTA-Cls | Claude | Accuracy | 0.33 | 25% | 0.33 | 25% | 0.73 |
| ***Image Quality & Enhancement Tasks*** | | | | | | | |
| LDCT-IQA | GPT | Score | 2.62 | 33% | 2.50 | 0% | 2.74 |
| LDCT-IQA | Gemini | Score | 2.70 | 83% | 2.62 | 17% | 2.74 |
| LDCT-IQA | Claude | Score | 1.12 | 0% | FAIL | – | 2.74 |

domain expertise. They lack evaluation of: (1) real competition grades, (2) actual model outputs, (3) domain expert requirements, and (4) the agent's strategy. Most importantly, they do not systematically analyze *why* agents fail or *what specific capabilities* are missing, questions essential for advancing agent development toward genuine domain expertise.

**AI Agents for Scientific Research.** The vision of autonomous AI agents for scientific research has gained significant momentum, with recent systems demonstrating the potential to autonomously explore solution spaces (Toledo et al., 2025), conduct end-to-end experiments (Team et al., 2025), and perform complex reasoning tasks. In machine learning engineering, agents like AIDE (Schmidt et al., 2024), ML-Master (Liu et al., 2025), and AIRA (Toledo et al., 2025) employ tree search strategies to tackle Kaggle competitions, while R&D-Agent (Yang et al., 2025b) achieves open-source state-of-the-art 35.1% medal rate on MLE-bench through a modular framework separating idea generation from implementation. Beyond machine learning, agents have shown promise in automated scientific discovery (Team et al., 2025), formulating hypotheses and designing experiments. However, a critical question remains: while these agents excel at general-purpose tasks, do they possess the specialized domain knowledge required for expert-level scientific work? Our work addresses this question by evaluating agents on medical imaging challenges that demand deep domain expertise, revealing fundamental limitations in current systems' ability to apply specialized knowledge.

**Medical AI and Evaluation.** The medical domain presents unique challenges for AI evaluation due to the critical importance of domain knowledge and specialized expertise. While traditional benchmarks like MedQA (Jin et al., 2021) have reached saturation with

models exceeding 90% accuracy (Zuo et al., 2025), they suffer from limited clinical relevance and lack of construct validity (Alaa et al., 2025), prompting expert-level benchmarks like MedXpertQA (Zuo et al., 2025). Medical imaging competitions on platforms like Grand Challenge provide more realistic evaluation, attracting 200+ participants. Eisenmann et al. (Eisenmann et al., 2023) conducted comprehensive analysis of biomedical competition winners, identifying 20 winning strategies including domain knowledge application, expert collaboration, data curation, and specialized preprocessing. However, a significant gap remains between the capabilities of winning humans and current MLE agents in addressing these challenges. Our work systematically evaluates agents on Grand Challenge competitions with realistic settings to analyze not just *whether* agents fail but *why* they fail and *what specific domain capabilities* they lack, revealing fundamental limitations in current agents' ability to acquire and apply domain expertise, insights critical for developing the next generation of domain-aware autonomous systems.

## 5. Discussion

**Domain Knowledge.** Deep learning for medical imaging requires not only machine learning expertise but also deep understanding of the underlying data. Analyses of winning competition solutions show that substantial medical domain knowledge is applied during data preprocessing, a time-consuming stage demanding visualization, parameter tuning, and strategic decision-making. In domains such as CT, automated deep learning–based windowing methods have only recently begun to see adoption (Zhang et al., 2025; Lee et al., 2018). These challenges highlight why agents struggle: success in medical imaging depends on nuanced, modality-specific judgments that emerge from years of experience, not simply from applying generic machine learning workflows. Without the ability to iteratively explore the data, validate assumptions, and adjust pipelines, agents fall short of the careful engineering that expert practitioners rely on. Although medical imaging knowledge is abundant in papers, forums, and open-source code, current agents struggle to leverage it effectively. Even widely adopted frameworks like nnU-Net (Isensee et al., 2020), with over 5,000 citations, are often overlooked because agents cannot judge when they are relevant, creating a retrieval-relevance gap where agents find methods but lack understanding to assess whether the task is truly 3D, whether the data support that approach, or whether computational resources are sufficient.

**Beyond Domain Knowledge: The Medical Engineering Gap.** The agents we evaluated (AIDE, R&D Agent, and ML-Master) lack the medical deep learning knowledge and engineering skills necessary for medical imaging solutions. Even with winning strategies in Table 2, they cannot replicate results as they are fundamentally designed for general machine learning tasks and lack critical infrastructure for specialized formats (DICOM, NIfTI) and domain libraries (nnU-Net, MONAI, SimpleITK). Their tree search and reasoning mechanisms, effective for tabular or natural image tasks, fail to navigate 3D volumetric complexities like patch-based training, variable voxel spacing, or clinical metrics (Dice coefficients, Hausdorff distances), producing code that either fails or trains clinically meaningless models. M$^3$Builder (Feng et al., 2025) addresses this through medical-specific templates but remains limited by predefined frameworks (nnU-Net, Transformers), with its M$^3$Bench success likely reflecting template-solvable problems rather than tacit engineering knowledge

for edge cases like unconventional protocols or debugging clinically implausible predictions without explicit error signals.

Additional failure modes outside of the winning strategies are based on the overall utility of the agents. For a given 80GB NVIDIA H100 GPU, the agents typically only utilized 10-20% of the full memory while training their models, representing a significant opportunity cost in computational efficiency that human practitioners would naturally optimize through larger batch sizes, model ensembles, or hyperparameter sweeps. This inefficiency is compounded by the fact that most of these agent architectures are inherently iterative in nature and can only work on one task at a time rather than multi-tasking as a human would. While human experts routinely run multiple experiments in parallel, monitor training curves across different model configurations, and contextualize results from concurrent trials, these agents follow strictly sequential workflows that dramatically extend development time and limit their ability to efficiently explore the solution space within fixed time budgets.

**Implications for Autonomous Scientific Research.** While this work focuses on medical imaging, the failure modes we identify (retrieval-relevance gaps, domain-specific engineering deficits, and inability to apply documented best practices) extend to other expert scientific domains (Schmidgall et al., 2024; Xie et al., 2024; Mitchener et al., 2025). Although agentic frameworks grounded in scientific insight have been developed (Ding et al., 2025; Jansen et al., 2024), current benchmarks primarily measure engineering competence rather than domain expertise. Our capability analysis in Figure 4 reveals that agents fail to demonstrate fundamental scientific practices identified in winning solutions: systematic failure case analysis, domain-appropriate preprocessing strategies, and metric-aligned model design. These competencies emerge from years of iterative experience, understanding which methods transfer across problem instances, recognizing data artifacts versus true signal, and knowing which engineering shortcuts compromise clinical validity. This gap has critical implications for autonomous agents in high-stakes domains like drug discovery, materials science, and genomics, where incorrect modeling decisions waste experimental resources and may impact patient safety. Current agents' inability to recognize when they lack domain knowledge, evident in confident but meaningless submissions, suggests they cannot reliably self-assess their competence boundaries, a prerequisite for safe autonomous operation. Achieving genuine scientific autonomy requires architectural innovations beyond scaling: mechanisms for domain-specific reasoning rather than retrieval pattern-matching and competence self-awareness to identify when human expertise is required.

**Bridging the Gap.** Several actionable solutions could address the current limitations of autonomous ML agents in medical imaging. First, developing domain-specialized agent architectures with native support for medical formats (DICOM/NIfTI) and pre-integrated frameworks (nnU-Net, MONAI) would reduce the engineering burden of handling specialized data. Second, implementing parallel experimentation capabilities would allow agents to run multiple configurations simultaneously, better utilizing GPU resources (currently only 10-20% utilized) and mirroring human expert workflows. Third, incorporating competence self-assessment mechanisms that flag unfamiliar modalities or request human verification would prevent confident but meaningless submissions. Fourth, enhancing retrieval-relevance systems with medical imaging-specific semantic understanding would help agents judge method applicability beyond simple pattern matching. Finally, domain-specific fine-tuning

on medical imaging literature and competition solutions could help foundation models develop the tacit engineering knowledge currently accessible only through years of human experience.

**Limitations.** Our work has several limitations. First, the standardized computational resources (H100 GPUs, 24-hour time budget) may not reflect all research environments. Second, potential data leakage exists if foundation models were pre-trained on competition solutions, though such knowledge did not translate to successful implementations. Third, our automated capability assessment provides scalability but lacks the depth of manual expert analysis, which would be prohibitively labor-intensive. Finally, ablation studies (Tables 4 & 5) evaluated only 4 representative challenges due to computational constraints.

## 6. Conclusion

In this paper, we introduced ReX-MLE, a 20-challenge medical imaging benchmark that exposes a substantial gap between general-purpose autonomous ML agents and the domain expertise required for medical AI. We showed that leading agents achieve only 4.53–12.15% mean percentile rank on ReX-MLE, with most submissions falling to the 0th percentile. Our capability analysis shows that these failures arise from missing scientific and engineering practices rather than insufficient time or model scale. Agents rarely demonstrate the core strategies used by human competition winners, even when provided with solution reports. These results challenge the notion that scaling general agents will naturally yield scientific competence and highlight the need for architectures that integrate domain knowledge, structured reasoning, and robust experimental workflows. We release ReX-MLE to support systematic progress toward autonomous systems capable of credible medical imaging research.

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

# Appendix A. Detailed Benchmark Results

Table S1: Agent performance across DENTEX challenge.

| Metric | AIDE | RD-Agent | ML-Master | Human |
|---|---|---|---|---|
| AP Mean | 0.0892 (11) | 0.0000 (11) | 0.0855 (11) | 0.3995 (1) |
| AP50 Mean | 0.1635 (11) | 0.0000 (11) | 0.1547 (11) | 0.5775 (5) |
| AP75 Mean | 0.0885 (11) | 0.0000 (11) | 0.0806 (11) | 0.4843 (1) |
| AR Mean | 0.4437 (11) | 0.0000 (11) | 0.2164 (11) | 0.6760 (4) |
| AP Quadrant | 0.1651 (11) | 0.0000 (11) | 0.1984 (11) | 0.4745 (1) |
| AP50 Quadrant | 0.3117 (11) | 0.0000 (11) | 0.3589 (11) | 0.6787 (3) |
| AP75 Quadrant | 0.1593 (11) | 0.0000 (11) | 0.1903 (11) | 0.5846 (1) |
| AR Quadrant | 0.5957 (11) | 0.0000 (11) | 0.4073 (11) | 0.7539 (4) |
| AP Enumeration | 0.0538 (11) | 0.0000 (11) | 0.0321 (11) | 0.3535 (1) |
| AP50 Enumeration | 0.0976 (11) | 0.0000 (11) | 0.0573 (11) | 0.5106 (1) |
| AP75 Enumeration | 0.0522 (11) | 0.0000 (11) | 0.0291 (11) | 0.4207 (1) |
| AR Enumeration | 0.3708 (11) | 0.0000 (11) | 0.1250 (11) | 0.6819 (4) |
| AP Diagnosis | 0.0486 (11) | 0.0000 (11) | 0.0259 (11) | 0.3706 (5) |
| AP50 Diagnosis | 0.0814 (11) | 0.0000 (11) | 0.0480 (11) | 0.5431 (7) |
| AP75 Diagnosis | 0.0542 (11) | 0.0000 (11) | 0.0225 (11) | 0.4477 (5) |
| AR Diagnosis | 0.3647 (11) | 0.0000 (11) | 0.1169 (11) | 0.6760 (4) |
| Mean Rank | 11.0 | 11.0 | 11.0 | 3.0 |
| Percentile | 0.0% | 0.0% | 0.0% | 100.0% |

Table S2: Agent performance across ISLES'22 challenge.

| Metric | AIDE | RD-Agent | ML-Master | Human |
|---|---|---|---|---|
| Dice | 0.0418 (11) | 0.0165 (11) | 0.0000 (11) | 0.7852 (2) |
| Lesion F1 | 0.0510 (11) | 0.0598 (11) | 0.0000 (11) | 0.8196 (2) |
| Lesion Count Difference | 59.4600 (11) | 9.7800 (11) | 10.8400 (11) | 2.1200 (4) |
| Absolute Volume Difference | 24.3939 (11) | 12.3190 (11) | 17.5051 (11) | 5.0713 (6) |
| Mean Rank | 11.0 | 11.0 | 11.0 | 3.5 |
| Percentile | 0.0% | 0.0% | 0.0% | 100.0% |
| Overall | 11.0 | 11.0 | 11.0 | 2.0 |

Table S3: Agent performance across LDCT-IQA challenge.

| Metric | AIDE | RD-Agent | ML-Master | Human |
|---|---|---|---|---|
| PLCC | 0.9244 | 0.9404 | 0.8909 | N/A |
| SROCC | 0.9243 | 0.9358 | 0.8860 | N/A |
| KROCC | 0.7761 | 0.7883 | 0.7214 | N/A |
| Score | 2.6248 (5) | 2.6645 (4) | 2.4983 (7) | 2.7427 (1) |
| Mean Rank | 5.0 | 4.0 | 7.0 | 1.0 |
| Percentile | 33.3% | 50.0% | 0.0% | 100.0% |

Table S4: Agent performance across PANTHER Task 1 challenge.

| Metric | AIDE | RD-Agent | ML-Master | Human |
|---|---|---|---|---|
| DSC | 0.3301 (11) | 0.1628 (11) | 0.1332 (11) | 0.7265 (1) |
| MSD | 0.4593 (11) | 0.2066 (11) | 0.1783 (11) | 0.9204 (1) |
| HD95 | 33.3998 (7) | 88.0805 (11) | 93.5596 (11) | 8.5993 (1) |
| MASD | 18.9004 (7) | 20.6307 (7) | 22.9698 (7) | 1.6730 (1) |
| RMSE | 25134.9312 (11) | 72218.6646 (11) | 81363.6640 (11) | 9338.0637 (1) |
| Mean Rank | 9.4 | 10.2 | 10.2 | 1.0 |
| Percentile | 16.0% | 8.0% | 8.0% | 100.0% |

Table S5: Agent performance across PANTHER Task 2 challenge.

| Metric | AIDE | RD-Agent | ML-Master | Human |
|---|---|---|---|---|
| DSC | 0.0950 (10) | 0.2844 (9) | 0.0495 (10) | 0.5289 (1) |
| MSD | 0.1363 (10) | 0.4458 (9) | 0.0682 (10) | 0.6999 (1) |
| HD95 | 103.8332 (10) | 21.7672 (1) | 305.5815 (10) | 23.0110 (1) |
| MASD | 31.3357 (9) | 9.3905 (6) | 284.1167 (10) | 5.1319 (1) |
| RMSE | 131723.8810 (11) | 12778.1561 (1) | 111239.35802 (11) | 17163.5753 (4) |
| Mean Rank | 10.0 | 5.2 | 10.2 | 1.6 |
| Percentile | 10.0% | 58.0% | 8.0% | 100.0% |

Table S6: Agent performance across PUMA-T1-Seg challenge.

| Metric | AIDE | RD-Agent | ML-Master | Human |
|---|---|---|---|---|
| Dice | FAIL | 0.0000 (11) | 0.0000 (11) | 0.7832 (1) |
| Mean Rank | FAIL | 11.0 | 11.0 | 1.0 |
| Percentile | FAIL | 0.0% | 0.0% | 100.0% |

Table S7: Agent performance across PUMA-T1-Det challenge.

| Metric | AIDE | RD-Agent | ML-Master | Human |
|---|---|---|---|---|
| Macro F1 | 0.0179 (11) | 0.0568 (11) | 0.0843 (11) | 0.6585 (1) |
| F1 Other | 0.0537 | 0.0186 | 0.0256 | N/A |
| F1 TILs | 0.0000 | 0.0000 | 0.0000 | N/A |
| F1 Tumor | 0.0000 | 0.1519 | 0.2273 | N/A |
| Mean Rank | 11.0 | 11.0 | 11.0 | 1.0 |
| Percentile | 0.0% | 0.0% | 0.0% | 100.0% |

Table S8: Agent performance across PUMA-T2-Seg challenge.

| Metric | AIDE | RD-Agent | ML-Master | Human |
|---|---|---|---|---|
| Dice | 0.0000 (11) | 0.0000 (11) | 0.0000 (11) | 0.7823 (1) |
| Mean Rank | 11.0 | 11.0 | 11.0 | 1.0 |
| Percentile | 0.0% | 0.0% | 0.0% | 100.0% |

Table S9: Agent performance across PUMA-T2-Det challenge.

| Metric | AIDE | RD-Agent | ML-Master | Human |
|---|---|---|---|---|
| F1 | FAIL | 0.0130 (11) | 0.0007 (11) | 0.2707 (1) |
| F1 Epithelium | FAIL | 0.0072 | 0.0000 | N/A |
| F1 Lymphocytes | FAIL | 0.0000 | 0.0000 | N/A |
| F1 Histiocytes | FAIL | 0.0000 | 0.0000 | N/A |
| F1 Tumor | FAIL | 0.0000 | 0.0000 | N/A |
| F1 Melanophages | FAIL | 0.0000 | 0.0000 | N/A |
| F1 Stromal Cells | FAIL | 0.0000 | 0.0000 | N/A |
| F1 Neutrophils | FAIL | 0.0000 | 0.0000 | N/A |
| F1 Plasma Cells | FAIL | 0.0000 | 0.0000 | N/A |
| F1 Apoptotic Cells | FAIL | 0.0000 | 0.0000 | N/A |
| F1 Endothelium | FAIL | 0.0000 | 0.0000 | N/A |
| Mean Rank | FAIL | 11.0 | 11.0 | 1.0 |
| Percentile | FAIL | 0.0% | 0.0% | 100.0% |

Table S10: Agent performance across SEG.A challenge.

| Metric | AIDE | RD-Agent | ML-Master | Human |
|---|---|---|---|---|
| HD 50th Percentile | 354.3906 (11) | 828.4886 (11) | 168.7597 (11) | 2.6125 (1) |
| DSC 50th Percentile | 0.0154 (11) | 0.0000 (11) | 0.0204 (11) | 0.9234 (2) |
| Mean Rank | 11.0 | 11.0 | 11.0 | 1.5 |
| Percentile | 0.0% | 0.0% | 0.0% | 100.0% |

Table S11: Agent performance across TopBrain-CTA challenge.

| Metric | AIDE | RD-Agent | ML-Master | Human |
|---|---|---|---|---|
| Dice | 0.0289 (11) | 0.0767 (11) | 0.2591 (11) | 0.7910 (1) |
| clDice | 0.0219 (11) | 0.0476 (11) | 0.3204 (11) | 0.8330 (1) |
| B0 Error | 22.7525 (11) | 38.4442 (11) | 5.6267 (10) | 0.7680 (2) |
| HD95 | 236.2618 (11) | 242.6999 (11) | 86.5582 (11) | 19.7770 (2) |
| Neighbor Error | 0.7863 (10) | 1.2344 (10) | 1.5962 (10) | 0.0000 (1) |
| F1 Side Road | 0.0000 (11) | 0.0000 (11) | 0.1290 (11) | 0.6780 (1) |
| Mean Rank | 10.83 | 10.83 | 10.67 | 1.33 |
| Percentile | 1.7% | 1.7% | 3.3% | 100.0% |

Table S12: Agent performance across TopBrain-MRA challenge.

| Metric | AIDE | RD-Agent | ML-Master | Human |
|---|---|---|---|---|
| Dice | 0.0138 (11) | 0.5016 (11) | 0.2631 (11) | 0.8140 (2) |
| clDice | 0.0172 (11) | 0.5686 (11) | 0.2422 (11) | 0.8630 (1) |
| B0 Error | 2.8767 (10) | 1097.5648 (11) | 35.1468 (11) | 0.7650 (2) |
| HD95 | 275.2090 (11) | 43.8033 (11) | 94.4634 (11) | 13.7830 (1) |
| Neighbor Error | 0.1107 (6) | 6.9057 (11) | 6.4116 (11) | 0.0000 (1) |
| F1 Side Road | 0.0000 (11) | 0.5009 (11) | 0.2234 (11) | 0.8530 (1) |
| Mean Rank | 10.0 | 11.0 | 11.0 | 1.33 |
| Percentile | 10.0% | 0.0% | 0.0% | 100.0% |

Table S13: Agent performance across TopCoW-CTA-Seg challenge.

| Metric | AIDE | RD-Agent | ML-Master | Human |
|---|---|---|---|---|
| Dice | 0.0865 (11) | 0.4933 (11) | 0.2463 (11) | 0.8700 (2) |
| clDice | 0.1997 (11) | 0.6769 (11) | 0.3999 (11) | 0.9900 (1) |
| B0 Error | 95.0882 (11) | 5.2585 (11) | 1.0344 (11) | 0.0400 (1) |
| HD95 | 84.7932 (11) | 43.0855 (11) | 40.4297 (11) | 3.2200 (3) |
| F1 GRP2 | 0.0000 (11) | 0.2217 (11) | 0.0000 (11) | 0.8600 (1) |
| Anterior Graph Accuracy | 0.0000 (11) | 0.3600 (11) | 0.0000 (11) | 0.8700 (3) |
| Posterior Graph Accuracy | 0.0000 (11) | 0.0400 (11) | 0.0000 (11) | 0.6800 (4) |
| Anterior Topology | 0.0000 (11) | 0.3600 (9) | 0.0000 (11) | 0.7900 (1) |
| Posterior Topology | 0.0000 (11) | 0.0400 (11) | 0.0000 (11) | 0.5800 (3) |
| Mean Rank | 11.0 | 10.78 | 11.0 | 2.11 |
| Percentile | 0.0% | 2.2% | 0.0% | 100.0% |

Table S14: Agent performance across TopCoW-CTA-Det challenge.

| Metric | AIDE | RD-Agent | ML-Master | Human |
|---|---|---|---|---|
| Boundary IoU | 0.6110 (5) | 0.6518 (2) | 0.5873 (7) | 0.6900 (1) |
| IoU | 0.6707 (7) | 0.6999 (7) | 0.6485 (7) | 0.7900 (1) |
| Mean Rank | 6.0 | 4.5 | 7.0 | 1.0 |
| Percentile | 37.5% | 56.3% | 25.0% | 100.0% |

Table S15: Agent performance across TopCoW-CTA-Cls challenge.

| Metric | AIDE | RD-Agent | ML-Master | Human |
|---|---|---|---|---|
| Anterior Accuracy | 0.3333 (4) | 0.2778 (4) | 0.1019 (7) | 0.7300 (2) |
| Posterior Accuracy | 0.1667 (6) | 0.1917 (4) | 0.0875 (7) | 0.8700 (1) |
| Mean Rank | 5.0 | 4.0 | 7.0 | 1.5 |
| Percentile | 33.3% | 50.0% | 0.0% | 100.0% |

Table S16: Agent performance across TopCoW-MRA-Seg challenge.

| Metric | AIDE | RD-Agent | ML-Master | Human |
|---|---|---|---|---|
| Dice | 0.1146 (11) | 0.7284 (11) | 0.4791 (11) | 0.8800 (4) |
| clDice | 0.0610 (11) | 0.7903 (11) | 0.5983 (11) | 0.9900 (1) |
| B0 Error | 229.7508 (11) | 5.4828 (11) | 9.9689 (11) | 0.0500 (2) |
| HD95 | 91.1558 (11) | 25.2004 (11) | 39.5812 (11) | 1.5000 (1) |
| F1 GRP2 | 0.0000 (11) | 0.5898 (10) | 0.1071 (11) | 0.9200 (1) |
| Anterior Graph Accuracy | 0.0000 (11) | 0.0800 (11) | 0.0000 (11) | 0.8900 (2) |
| Posterior Graph Accuracy | 0.0000 (11) | 0.2800 (11) | 0.0000 (11) | 0.7700 (2) |
| Anterior Topology | 0.0000 (11) | 0.0800 (10) | 0.0000 (11) | 0.6000 (1) |
| Posterior Topology | 0.0000 (11) | 0.2800 (10) | 0.0000 (11) | 0.6000 (2) |
| Mean Rank | 11.0 | 10.67 | 11.0 | 1.78 |
| Percentile | 0.0% | 3.3% | 0.0% | 100.0% |

Table S17: Agent performance across TopCoW-MRA-Det challenge.

| Metric | AIDE | RD-Agent | ML-Master | Human |
|---|---|---|---|---|
| Boundary IoU | 0.5993 (7) | 0.1126 (7) | 0.6392 (7) | 0.7700 (1) |
| IoU | 0.6587 (7) | 0.1867 (7) | 0.6893 (7) | 0.8500 (1) |
| Mean Rank | 7.0 | 7.0 | 7.0 | 1.0 |
| Percentile | 14.3% | 14.3% | 14.3% | 100.0% |

Table S18: Agent performance across TopCoW-MRA-Cls challenge.

| Metric | AIDE | RD-Agent | ML-Master | Human |
|---|---|---|---|---|
| Anterior Accuracy | 0.3333 (4) | 0.0926 (7) | 0.3333 (4) | 0.8900 (1) |
| Posterior Accuracy | 0.0714 (7) | 0.0556 (7) | 0.1698 (7) | 0.7500 (1) |
| Mean Rank | 5.5 | 7.0 | 5.5 | 1.0 |
| Percentile | 25.0% | 0.0% | 25.0% | 100.0% |

Table S19: Agent performance across USEnhance challenge.

| Metric | AIDE | RD-Agent | ML-Master | Human |
|---|---|---|---|---|
| LNCC | 0.1092 (11) | FAIL | 0.1295 (11) | 0.9080 (1) |
| SSIM | 0.2866 (11) | FAIL | 0.3219 (11) | 0.7439 (2) |
| PSNR | 15.8771 (11) | FAIL | 16.1628 (11) | 30.7268 (1) |
| Mean Rank | 11.0 | FAIL | 11.0 | 1.33 |
| Percentile | 0.0% | FAIL | 0.0% | 100.0% |

Table S20: Agent performance across NeurIPS-CellSeg challenge.

| Metric | AIDE | RD-Agent | ML-Master | Human |
|---|---|---|---|---|
| F1 @ 0.5 | 0.0362 (11) | 0.3631 (11) | 0.0364 (11) | 0.8770 (1) |
| F1 @ 0.6 | 0.0189 (11) | 0.3232 (11) | 0.0263 (11) | 0.8464 (1) |
| F1 @ 0.7 | 0.0077 (11) | 0.2769 (11) | 0.0189 (11) | 0.8051 (1) |
| F1 @ 0.8 | 0.0023 (11) | 0.1979 (11) | 0.0125 (11) | 0.7048 (1) |
| F1 @ 0.9 | 0.0001 (11) | 0.0779 (11) | 0.0039 (11) | 0.3901 (3) |
| Mean Rank | 11.0 | 11.0 | 11.0 | 1.4 |
| Percentile | 0.0% | 0.0% | 0.0% | 100.0% |

## Appendix B. Automated Capability Analysis Pipeline

To investigate the underlying causes of the performance gaps observed in Medical MLE-Bench, we conducted a structured analysis of the process-level behavior of each agent. Rather than relying solely on outcome metrics, we examined the execution traces generated during each challenge, following the 13 "Winning Strategies" identified by Eisenmann et al. (Eisenmann et al., 2023). Given the size of our benchmark, 60 execution traces across 20 challenges, manual annotation was impractical. We therefore developed an automated LLM-as-a-judge pipeline powered by a state-of-the-art large language model (GPT-5), which evaluates each strategy in a consistent, reproducible manner.

### B.1. Overview of the LLM-as-a-judge Procedure

For every agent, challenge pair, the LLM receives the full execution log, including: natural-language reasoning traces produced by the agent, shell commands and their outputs, Python code snippets and error messages, training logs, and intermediate analyses. The LLM is instructed to assign a binary score for each of the 13 strategies: **1** if there is *explicit, verifiable evidence* that the agent implemented the strategy and **0** if the evidence is ambiguous, missing, or only planned but not executed.

This evaluation produces a structured record of observed capabilities, which we aggregate across all tasks to produce the capability profiles shown in Figure 4. The method ensures high consistency across agents and challenges while isolating the specific scientific practices that current autonomous ML agents fail to use in practice. Below we provide the exact template used in our implementation.

### B.2. System Prompt

```
You are an expert adjudicator evaluating the execution logs of an
autonomous AI agent on a medical imaging task. Your objective is to
determine whether the agent implemented specific technical strategies
based on explicit evidence in the logs.

INSTRUCTIONS:
1. Review the provided Log Content deeply.
2. For EACH Strategy Definition, decide if there is explicit evidence
   of execution.
3. Score 1 ONLY if there is explicit evidence of execution (code execution,
   specific library calls, distinct file outputs).
4. Score 0 if the strategy is ambiguous, merely planned but not executed,
   or absent.
5. Return a raw JSON object (no Markdown) with a 'results' list of objects
   containing 'id', 'strategy', 'score', and 'evidence'.

Output Format:
{"results":[{"id":1, "strategy":"...", "score":0, "evidence":"..."}]}
```

**B.3. Strategy Prompt Structure**

Each strategy is defined in a JSON file as an object with three fields:

```
{
  "id": <integer>,
  "name": "<strategy name>",
  "criteria": "<description of explicit evidence required>"
}
```

During evaluation, each strategy is rendered into the following prompt block:

```
TARGET STRATEGY: <name>
<criteria>
```

The LLM receives all 13 strategies concatenated into a single message, followed by:

```
Log section (<label>):
<log content>

Return JSON with key 'results' containing one object per strategy.
```

**B.4. List of Strategies Used**

Below we include the complete strategy list used in our evaluation, along with the corresponding criteria defining what constitutes explicit evidence of implementation.

1. **Analyzing and handling failure cases**
   Explicit evidence includes inspection of errors or bad predictions (stack traces, broken samples, metric failures), identification of root causes, and application of code or data fixes to resolve them.

2. **Knowing the state of the art**
   References to, or usage of, medical-imaging–specific state-of-the-art architectures, benchmarks, or literature (e.g., UNet variants, SAM, Swin Transformers), going beyond default or generic ML models.

3. **Reflecting metrics in method design**
   Evidence that losses, thresholds, or postprocessing are explicitly tailored to the target metric (e.g., Dice/IoU), including metric-aware tuning or threshold sweeps.

4. **Having domain knowledge**
   Use of modality-specific preprocessing or reasoning steps (e.g., HU windowing for CT, isotropic resampling, spacing or anisotropy correction, organ-specific priors).

5. **Rapid experiment iteration pipeline**
   Creation of automation for fast iteration (e.g., structured configs, training scripts, logging, checkpointing, or hyperparameter sweeps) as opposed to ad-hoc single runs.

6. **Optimizing the augmentation method**
Use or tuning of augmentation frameworks (e.g., Albumentations, MONAI), ablations of augmentation choices, or explicit optimization of augmentation parameters.

7. **Incorporating domain expert priors**
Inclusion of expert-inspired rules or anatomical/clinical heuristics (e.g., viable shape constraints, plausible value ranges, organ-specific filtering) in training or postprocessing.

8. **Data curation and cleaning**
Evidence of detecting and repairing problematic data (e.g., corrupt or missing files, mismatched labels, inconsistent headers, class imbalance handling, filtering).

9. **Postprocessing results**
Explicit postprocessing applied to predictions, such as connected-component filtering, morphological operations, hole filling, box/score filtering, or test-time augmentation fusion.

10. **Ensembling heterogeneous models**
Combining multiple different architectures or checkpoints into a unified prediction (e.g., averaging, weighted fusion, majority voting).

11. **Leveraging external data**
Use of datasets or pretrained weights beyond the provided training set (e.g., ImageNet, public domain medical data, pretrained segmentation backbones).

12. **Ensembling via seeds/folds**
Training multiple seeds or cross-validation folds and merging predictions during inference.

13. **Optimizing hyperparameters systematically**
Structured hyperparameter search using grid/random/Bayesian/Optuna-based sweeps, or scripted comparisons with logged results.

