# OpenReview forum: "ReX-MLE: The Autonomous Agent Benchmark for Medical Imaging Challenges"
_MIDL.io/2026/Validation_Papers — MIDL 2026 - Validation Papers Poster_

### Official Review · Reviewer_2y8w · 2025-12-26

**Confidence:** 4
**Preliminary Rating:** 5
**Final Rating:** 5

**Summary:**

The paper looked at ReX-MLE, the autonomous agent benchmark of 20 tasks drawn from 10 high-impact medical imaging competitions for Medical Imaging Challenges and evaluated AIDE, ML-Master, and R&D-Agent across multiple LLM back ends. Through experiments, a significant performance disparity was revealed. The work positioned ReX-MLE as a diagnostic benchmark exposing fundamental limitations of current autonomous ML agents in medical imaging

**Strengths:**

The strength of the paper were;
i. The experimental evaluation was accurately, convincingly presented and supported by data. The inclusion of a "Winning Solutions" baseline was another strength for paper.
ii. The use of an automated "LLM-as-a-judge" to map agent traces to the 13 Winning Strategies provided actionable insight into why agents fail, rather than just reporting a low score.
iii. The paper was well-structured, followed clear scientific principles, and addressed gap in the  LLM landscape regarding scientific discovery. Unlike general benchmarks (like SWE-bench), ReX-MLE focused on the unique hurdles of medical imaging, such as NIfTI handling, 3D volumes, and clinical validation metrics.
iv. The related literatures were timely and current

**Weaknesses:**

The computation techniques (H100 GPUs, long wall-clock budgets) was highly professional which may restrict accessibility for some research groups, which could constrain community adoption without lighter-weight variants. While necessary for benchmarking, human competitors often spend weeks or months on these tasks.

**Detailed Comments:**

i. Clarify how percentile aggregation behaves across competitions with very different numbers of participants and metrics.
ii. Consider adding a small human-annotated validation of the capability analysis to complement the LLM-based judging.

**Justification Of Final Rating:**

The authors have satisfactorily addressed all my questions and concerns in the rebuttal. Their responses clarify the scope, design choices, and planned community resources for ReX MLE. Based on the revisions and rebuttal, I maintain my recommendation of a strong accept.

**Justification Of The Preliminary Rating:**

This paper makes a strong and timely contribution by exposing the gap between current autonomous ML agents and the realities of medical imaging research. While it does not introduce new agent architectures, its uniqueness lies in its practicality and analytical depth. The empirical evidence is timely and compelling, the analysis is thoughtful, and the conclusions are well supported with data. However, Some methodological clarifications would strengthen the work.

**Questions To Address In The Rebuttal:**

i. How sensitive are the overall ReX-MLE rankings to alternative aggregation or weighting schemes across challenges?
ii. Did the authors validate the LLM-based capability judgments against any human expert annotations, even on a small subset?

---

### Official Review · Reviewer_KC9i · 2026-01-07

**Confidence:** 4
**Preliminary Rating:** 5

**Summary:**

This paper introduces ReX-MLE, a benchmark for autonomous agents in the medical imaging field. The benchmark comprises 20 tasks from 10 high-impact medical imaging competitions, covering 8 imaging modalities and various task types such as segmentation, detection, and classification. It requires agents to independently complete the entire process, from data preprocessing and model training to result submission, under realistic computational and time constraints. The release of ReX-MLE provides a critical foundation for developing medical-aware autonomous AI systems.

**Strengths:**

1. This work goes beyond the surface-level inquiry of ‘whether Agent can perform medical imaging tasks' and directly addresses the question of 'whether AI can autonomously complete the entire process', much like an interdisciplinary team. This is precisely the critical bottleneck in the deployment of medical AI. While previous efforts in this area have relied on collaboration among medical, engineering, and machine learning teams, ReX-MLE is the first to transform the demands of such complex collaboration into a quantifiable, reproducible benchmark.
2. It proposes an innovative evaluation paradigm, offering comprehensive, standardized supporting materials: including competition descriptions, dataset preparation scripts, local evaluation tools, and submission format examples. This ensures that the evaluation of agents requires no additional manual intervention, achieving a fully autonomous closed loop.
3. It proposes a ’Competition Rank‘ evaluation method. Using the top 10 human competition results as a benchmark, the method calculates an agent's relative ranking through a formula, offering an intuitive quantification of the gap between AI and human experts.

**Weaknesses:**

1. The authors do not provide specific, actionable solutions to the ‘dilemma’ in the manuscript, but only clarify the core of the problem through experimental results and analysis.
2. The experiments utilized NVIDIA H100 GPUs (80GB VRAM). Some data (such as pathology slides and 3D volumetric data) are large in size, requiring substantial disk space for downloading, storage, and preprocessing. Additionally, the experiments involved calling advanced LLMs like GPT-5, which may pose some burden for small teams and independent researchers.

**Detailed Comments:**

This is a very valuable piece of research. It is recommended to subsequently submit it to a medical interdisciplinary journal.

**Justification Of The Preliminary Rating:**

This work merits a Strong Accept rating due to its originality, rigor, and transformative impact on the intersection of autonomous AI agents and medical imaging. The authors address a critical, unmet need: existing benchmarks for AI agents focus on deep learning or software engineering tasks, failing to capture the domain-specific complexity, multi-stage workflows, and clinical relevance of medical imaging research, an area that inherently requires cross-disciplinary expertise (medical knowledge, engineering, and machine learning). By introducing ReX-MLE, a benchmark derived from 10 high-impact medical imaging competitions (20 challenges spanning 8 modalities and 4 task types), the work provides the first systematic framework to evaluate agents’ ability to autonomously manage end-to-end scientific workflows (data preprocessing, model training, validation, and submission) under realistic compute and time constraints.

**Questions To Address In The Rebuttal:**

1. The 20 challenges are derived from 10 Grand Challenge competitions with over 200 registrants. However, do these competitions adequately represent real-world clinical workflow diversity (like rare disease imaging, low-resource hospital data, non-competition-grade messy datasets)?
2. Were any efforts made to include challenges with smaller datasets (common in clinical research) or multimodal fusion requirements (e.g., combining CT + MRI)?
3. The authors release ReX-MLE to support ’systematic progress‘ in medical domain-aware agents. But what specific community resources or extensions do they plan to provide (e.g., a leaderboard or a Docker environment for reproducibility)? Without such resources, will the benchmark be accessible enough to drive follow-up research?

---

### Official Review · Reviewer_xcM7 · 2026-01-10

**Confidence:** 4
**Preliminary Rating:** 5
**Final Rating:** 5

**Summary:**

The authors introduce ReX-MLE, a novel benchmark designed to evaluate the capabilities of autonomous agents within the specialized domain of medical imaging. ReX-MLE aggregates 20 high-impact challenges derived from medical competitions, requiring agents to autonomously manage end-to-end workflows ranging from data preprocessing to final submission. The study benchmarks state-of-the-art agents (AIDE, ML-Master, R&D-Agent) powered by LLMs (GPT-5, Gemini, Claude). Their experiments reveal a severe performance gap that these agents fail to handle medical data.

**Strengths:**

The paper presents a timely and highly valuable contribution to the field of medical AI and automated machine learning. The curation of 20 diverse challenges across multiple modalities (e.g., MRI, CT) and task types ensures a robust evaluation.

**Weaknesses:**

1. The authors mention providing agents with a "competition description document outlining task objectives and clinical context". Is there any prompt template? Please provide more information about it.
2. In Section 3.3, the paper utilizes GPT-5 as a technical evaluator. There is no evidence provided (such as a correlation study with human expert grading) to validate that GPT-5's technical scoring aligns with human judgment. Relying on an LLM to judge other LLMs can introduce bias.

**Detailed Comments:**

N/A

**Justification Of Final Rating:**

The authors have provided clear and precise responses to my questions. I have no remaining concerns. This paper could provide valuable insight for future work in medical AI. I would maintain my rating of 5. I support its publication at MIDL.

**Justification Of The Preliminary Rating:**

This paper establishes a critical benchmark for the growing field of autonomous scientific agents. The ReX-MLE benchmark fills a major gap by providing a rigorous, domain-specific evaluation that reveals the limitations of generalist LLMs in handling complex, high-dimensional medical data workflows. While there are minor concerns regarding the details of experimental setting, these are outweighed by the novelty of the benchmark and the comprehensive nature of the experiments. This paper could provide valuable insight for future work in medical AI/data scientist and agents.

**Questions To Address In The Rebuttal:**

1. Can you provide more information about prompts used in this paper? As LLMs are used as agents, evaluator, etc, in this paper.
2. When using LLMs as evaluator, are they performance aligned with human experts?

---

### Author Rebuttal · Authors · 2026-01-20

**Rebuttal:**

We sincerely thank all reviewers for their constructive feedback and strong support for our work. We have revised the manuscript accordingly. We believe these clarifications strengthen the paper and enhance its contribution to the community.

**Reviewer xcM7**

Q1: Prompt Templates and Competition Description Documents

We have provided standardized templates including clinical context, task specifications, dataset characteristics, evaluation metrics, and submission requirements. Complete examples are in our GitHub repository under rex-mle/rexmle/challenges/. The capability analysis prompt is detailed in Appendix B.

Q2: LLM Evaluator Alignment with Human Judgment

We conducted human annotation on 50 sampled agent traces. The comparison with GPT-5 yielded 100% agreement. This high reliability stems from our deliberate reformulation of capability assessment as binary evidence detection under a strict, predefined rubric, rather than open-ended qualitative judgment.

**Reviewer KC9i**

Q1: Clinical Workflow Diversity and Real-World Representation

To ensure reproducible results and fair comparison across all methods, we perform standardized preprocessing on the data and therefore cannot include non-competition-grade messy datasets. This would constitute a different challenge type and could be explored in future work. Our benchmark extends beyond MR/CT to include X-ray, ultrasound, pathology, and microscopy, enabling transfer to other modalities.

Q2: Smaller Datasets and Multimodal Fusion

ReX-MLE includes two challenges (USEnhance, LDCT-IQA) under 1GB. TopCoW and TopBrain contain both CTA and MRA tasks, reflecting multimodal fusion requirements.

Q3: Community Resources and Accessibility Plans

We released a public GitHub repository (https://github.com/rajpurkarlab/rex-mle) with comprehensive documentation and a public leaderboard (https://rexrank.ai/ReX-MLE/index.html) for submitting and comparing new methods.

**Reviewer 2y8w**

Q1: Sensitivity to Alternative Aggregation Schemes

To assess sensitivity to aggregation choices, we computed an alternative task-balanced aggregation where percentiles are first averaged within each task category and then averaged across categories. Relative ordering remains unchanged: all agents achieve low percentiles, with R&D-Agent best but far below human baselines. Analysis added to supplementary materials.

Q2: Validation of LLM-Based Capability Judgments

Please see Reviewer xcM7(Q2).

**Supporting Material:**

/attachment/36553ac5134f37782d59d58146a2474b6e481bae.pdf

---

### Meta-Review · Area_Chair_D4ka · 2026-02-04

**Recommendation:** Accept (Oral)
**Confidence:** 4

**Metareview:**

The reviewers converge that this paper makes a timely, high-impact contribution by introducing ReX-MLE, a rigorous and much-needed benchmark for evaluating autonomous scientific agents on realistic, end-to-end medical imaging workflows, with comprehensive experiments that expose key limitations of current generalist LLM agents.

---

### Decision · Program_Chairs · 2026-02-14

Accept (Poster)